# Black carbon footprint of human presence in Antarctica

Raúl R. Cordero [1], Edgardo Sepúlveda[1], Sarah Feron[1,2 ✉], Alessandro Damiani[3 ✉], Francisco Fernandoy [4], Steven Neshyba[5], Penny M. Rowe[6], Valentina Asencio[7], Jorge Carrasco [8], Juan A. Alfonso[9], Pedro Llanillo [10], Paul Wachter [11], Gunther Seckmeyer[12], Marina Stepanova[1 ✉], Juan M. Carrera[1], Jose Jorquera[1], Chenghao Wang [13], Avni Malhotra [14], Jacob Dana[15], Alia L. Khan[15,16] & Gino Casassa[8]

Black carbon (BC) from fossil fuel and biomass combustion darkens the snow and makes it melt sooner. The BC footprint of research activities and tourism in Antarctica has likely increased as human presence in the continent has surged in recent decades. Here, we report on measurements of the BC concentration in snow samples from 28 sites across a transect of about 2,000 km from the northern tip of Antarctica (62°S) to the southern Ellsworth Mountains (79°S). Our surveys show that BC content in snow surrounding research facilities and popular shore tourist-landing sites is considerably above background levels measured elsewhere in the continent. The resulting radiative forcing is accelerating snow melting and shrinking the snowpack on BC-impacted areas on the Antarctic Peninsula and associated archipelagos by up to 23 mm water equivalent (w.e.) every summer.

[1] Universidad de Santiago de Chile. Av. Bernardo O'Higgins, 3363 Santiago, Chile. [2] University of Groningen, 8911 CE Leeuwarden, The Netherlands. [3] Center for Environmental Remote Sensing, Chiba University, 1-33 Yayoicho, Inage Ward, Chiba 263-8522, Japan. [4] Universidad Andrés Bello, Quillota 980, Viña del Mar, Chile. [5] University of Puget Sound, Department of Chemistry, Tacoma, WA, USA. [6] NorthWest Research Associates, Redmond, WA, USA. [7] Select Carbon Pty Ltd, 562 Wellington Street, Perth, WA 6000, Australia. [8] University of Magallanes, Av. Manuel Bulnes 1855, Punta Arenas, Chile. [9] Instituto Venezolano de Investigaciones Científicas (IVIC), Apartado, 20632 Caracas, Venezuela. [10] Alfred Wegener Institute (AWI), Am Handelshafen 12, 27570 Bremerhaven, Germany. [11] German Aerospace Center (DLR), German Remote Sensing Data Center (DFD), Wessling, Germany. [12] Leibniz Universität Hannover, Herrenhauser Strasse 2, Hannover, Germany. [13] Department of Earth System Science, Stanford University, Stanford, CA 94305, USA. [14] University of Zurich, Winterthurerstrasse 190, 8057 Zürich, Switzerland. [15] Western Washington University, 516 High St, Bellingham, WA 98225, USA. [16] National Snow and Ice Data Center, Cooperative Institute for Research in Environmental Sciences, University of Colorado—Boulder, Boulder, CO, USA. ✉email: s.c.feron@rug.nl; damiani@chiba-u.jp; marina.stepanova@usach.cl

Deposition of light-absorbing impurities (such as crustal dust and soot) onto snow-covered surfaces reduces the albedo, increases the fraction of solar energy absorbed, and accelerates melting[1]. Soot or black carbon (BC) is primarily produced during combustion in diesel engines, coal burning, wildfires, and residential wood burning[2] and may undergo regional and intercontinental transport during its short atmospheric lifetime. As a result, BC has been found (although at concentrations much lower than in source regions) in snow samples from remote regions in the Arctic[3–7], North America[8,9], Northern China[10], the Tibetan Plateau[11], the Nepalese Himalayas[12], and the Andes[13,14].

The presence of BC has been also confirmed in Antarctic snow[15–18] and in Antarctic ice cores[19–21]. These efforts have shown that background levels of BC in the Antarctic cryosphere, expressed as a concentration in mass per unit mass of meltwater extracted (e.g., ng/g), are consistently below 1 ng/g, at least one order of magnitude lower than in Arctic snow[3–7].

The remarkably low background concentrations of BC in Antarctica are compatible with simulations of long-range BC transport from prominent sources in the Southern Hemisphere[22,23] and with observations of episodic intercontinental transport of aerosols. Smoke from wildfires in South America[24] and in Australia[25,26] has been detected over Antarctica as well as dust originating in Central Australia[27] and in southern Patagonia[28]. However, back-trajectory analysis suggests that long-range transport plays only a minor role as a source of absorbing aerosols in Antarctica[29].

As limited meridional airborne transport restricts southward dispersal[30], local sources appear to dominate deposition of light-absorbing impurities in Antarctica. Ice-free areas in the northeast Antarctic Peninsula (about 312 km$^2$) drive dust deposition into that region[31,32], while BC content in snow surrounding research stations, such as Palmer Station[16], McMurdo Dry Valley Field Stations[17] and the Amundsen-Scott South Pole Station[18], has been shown to be considerably above background levels.

The BC footprint of local activities in Antarctica has likely increased as human presence in the continent has surged (Fig. 1). Vessels, airplanes, diesel power plants, generators, helicopters, and trucks are all local BC-rich sources that affect snow several kilometers downwind[16,18]. According to data from the Council of Managers of National Antarctic Programs (COMNAP)[33] and the Antarctic Treaty Secretariat (ATS)[34], 76 research stations are actively in use within the Antarctic Treaty area, providing accommodation for ca. 5500 people in summer. Some of these research facilities have developed into logistic hubs. Most stations are on or near the coast, and about half are located on the northern Antarctic Peninsula and associated archipelagos. The vast majority of tourist shore landings also occur in that region[35]. According to the International Association of Antarctica Tour Operators (IAATO)[36], ~74,000 tourists visited Antarctica in the 2019–2020 season, a 32% increase from the 2018–2019 season, and more than twice the total a decade ago. Of the visitors who travelled with IAATO members to Antarctica, most come by cruise ship but ~1% travelled by aircraft to destinations in the continental interior, such as Union Glacier Camp (79°S, Ellsworth Mountains).

Here, we report on a large-scale survey of the BC footprint of local anthropogenic activities (research and tourism) in Antarctica. Although we also sampled at two sites in the interior of the continent, we focused on the Antarctic Peninsula and associated archipelagos that, unlike the rest of the continent, have no large areas left untouched by humans. We measured the BC content in 155 snow samples collected in four consecutive summer seasons (from 2016–2017 to 2019–2020) at a total of 28 sites across a transect of about 2000 km from the northern tip of Antarctica (King George Island, 62°S) to the southern Ellsworth Mountains (Union Glacier Camp, 79°S); see Table 1 for further details.

## Results

**BC concentration.** We applied the Meltwater Filtration (MF) Technique[4–10,13,18] that required vacuum-filtering the sample meltwater. Filters were analyzed for their content of light-absorbing impurities by spectrophotometry. The spectrally resolved change in light transmission through the filters was then interpreted as an equivalent absorption by a certain mass of light-absorbing material per unit area of the filter. The equivalent absorption allowed us to assess the Ångström exponent of light-absorbing impurities in samples and, by apportioning the absorption to BC and non-BC components, to estimate the nanograms of BC per gram of snow (e.g., ng/g).

Since we aimed to assess the influence of light-absorbing impurities on the albedo of representative areas, sampling sites were chosen to be always hundreds of meters (when possible, several kilometers) away from apparent BC sources such as research stations, tourist shore landings, roads, and airfields (Supplementary Fig. 1). On the Antarctic Peninsula, our sampling expeditions roughly mirrored typical touristic routes. Most cruises visit the South Shetland Islands and the Palmer Archipelago also visiting small islands off the Antarctic Peninsula such as Cuverville (64°S), Petermann (65°S), and Detaille (67°S). We also sampled at the northern tip of the Antarctic Peninsula (63°S); specifically at Hope Bay (the site of the Argentinian Antarctic settlement Esperanza Base) and nearby the Chilean O'Higgins Station (about 30 km southwest of the northernmost point of the continent). In the Ellsworth Mountains, we sampled at Union Glacier (79°S), at 2 sites that are about 6 km East of a blue-ice runway and 1 km from the designated landing sites for ski-equipped airplanes.

The BC content found in our snow samples was considerably above background levels measured elsewhere in the continent[15–21]. The median of BC concentrations was about 3 ng/g (Fig. 2), which is close to the median of BC concentrations in snow samples collected in Greenland[5,6] but is still well below BC concentrations measured in any other distant region such as the Arctic[3–7], North America[8,9], Northern China[10], the Tibetan Plateau[11], the Nepalese Himalayas[12], and the Andes[13,14]. The BC content of snow in remote regions of the northern hemisphere is on the order of 20 ng/g[37].

A latitude dependence of BC concentration is apparent in Fig. 2b. On the northern South Shetland Islands (including King George Island and Greenwich Island) BC concentration generally ranged from 3 to 5 ng/g, while on the southern South Shetland Islands (including Livingston Island) and the northern tip of the Antarctic Peninsula (the Trinity Peninsula) BC concentration generally ranged from 4 to 7 ng/g. The highest levels of soot (about 8 ng/g) were measured 2 km southeast of the Argentinian Esperanza Base (63.4°S). BC decreased continuing south to the Palmer Archipelago; BC concentration generally ranged from 3 to 5 ng/g on Trinity Island (63.9°S) but was generally lower than 2 ng/g on Doumer Island (64.9°S). The low BC content in samples collected on Doumer Island (at sites close to the southernmost point of the island about 25 km east of the U.S. Palmer Station) approximately matches prior BC measurements conducted in the region[16]. BC concentrations further decreased southward along the Antarctic Peninsula to a minimum at the southernmost sampling sites; BC concentration generally ranged from 1 to 2 ng/g on Peterman Island (65.2°S) and Detaille Island (66.9°S), two popular tourist-landing sites but without permanent settlements.

**Dust presence.** A latitude dependence of absorption Ångström exponent ($\alpha$) is less clear (Fig. 2c). This exponent characterizes spectral absorption properties of light-absorbing impurities in snow. The two most common light-absorbing particles are soot and crustal dust. Under most conditions, BC dominates light

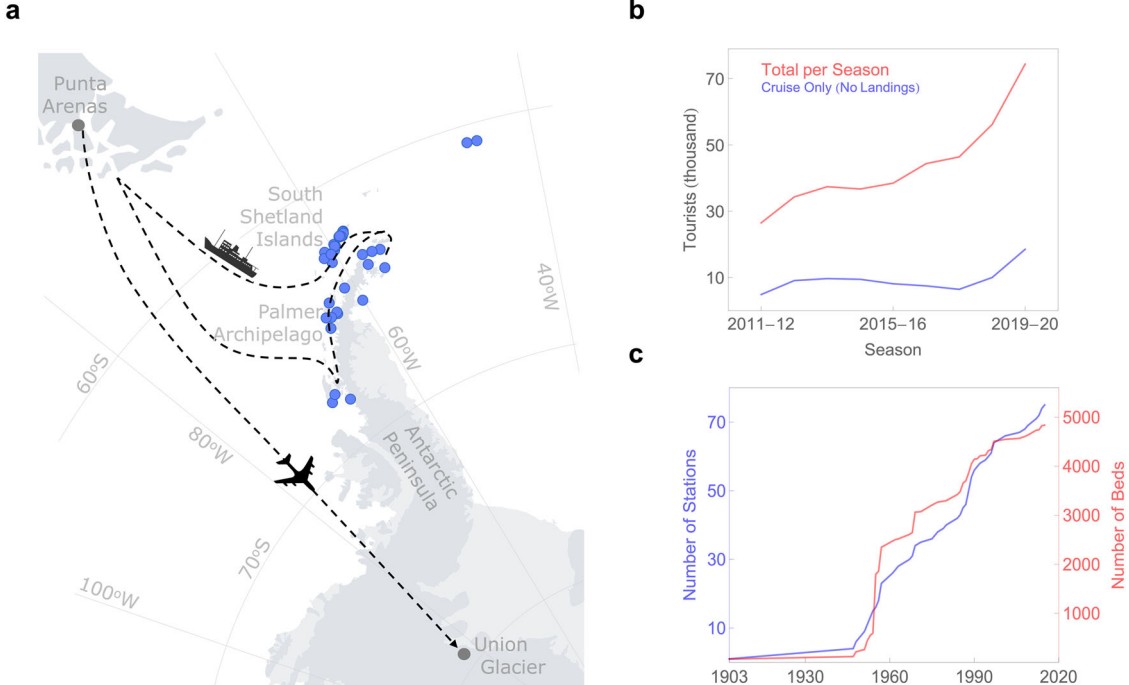

**Fig. 1 Human presence in Antarctica has surged in recent decades. a** Blue dots represent research stations in the area of interest according to the Council of Managers of National Antarctic Programs (COMNAP)[33]. Dotted lines represent popular tourist routes. Most of the visitors travelling to Antarctica with International Association of Antarctica Tour Operators (IAATO) embark on a ship for cruising the Antarctic Peninsula, where about half of the research facilities in the continent are located. Around 1% of all visitors fly to the interior of Antarctica where they stay in field camps such as Union Glacier Camp. IAATO has now more than 50 operators whose fleet of 54 vessels (including 6 large cruise ships) registered a total of 378 departures in the 2019–2020 season[36]. **b** Visitors have been growing steadily since 2011–2012. Of the 74,401 visitors who travelled with IAATO members to the region, about 25% travelled on cruise-only vessels (i.e., vessels carrying more than 500 passengers) and did not set foot on the continent. **c** There are currently 76 research stations in Antarctica (including seasonal facilities) with a combined accommodation capacity (for both scientists and staff) of about five thousand beds[33]. There are 11 research stations (with a total of 700 beds) on King George Island only[33]. Plots were generated by using Python's Matplotlib Library[58].

absorption since it has a mass absorption coefficient (MAC) that is several orders of magnitude greater than dust at visible wavelength[18]. For samples that contain both BC and dust, the Ångström exponent typically ranges from 1.1 (corresponding to pure BC) to about 4 (corresponding to dust)[10]. How much $\alpha$ values deviate from 1.1 is an indication of the relative contribution of dust to light absorption. The fact that the absorption Ångström exponent of light-absorbing impurities in our samples ranged from 2 to 3 highlights the contribution of non-BC constituents to light absorption, especially on the Antarctic Peninsula. Most of the extensive Byers Peninsula (about 60 km²), forming the west extremity of Livingston Island (South Shetland Islands), is ice-free in summer; the same goes for the Ulu Peninsula (about 312 km²) in the northeastern Antarctic Peninsula region[31]. Ice-free areas in Antarctica have been shown to be an important source of dust[31,32].

Statistical significance tests (Supplementary Fig. 2, Supplementary Tables 1–2) confirm that measurements of the BC concentration and the Ångström exponent at sites on the Antarctica Peninsula and associated archipelagos are significantly different from those taken at sites on Union Glacier (in the vast interior of the continent, 79°S). Yet, even at these remote locations, both BC and dust deposition are present (Fig. 3). Union Glacier is the site of a blue-ice runway and two seasonally occupied camps (a Chilean camp and another operated by a private company). In addition to the blue-ice runway, both camps operate ski-equipped airplanes. These facilities dominate BC deposition in this region, while the ice-free summits of the surrounding Ellsworth Mountains are likely the source of most of

the dust found in local snow. In samples taken from the surface, 6 km east of the blue-ice runway, BC concentration generally ranged from 1 to 3 ng/g. These concentrations likely resulted from the fact that winds at Union Glacier predominantly come from the southwest, and roughly match prior BC measurements conducted downwind from the Amundsen-Scott station and its airfield[18]. The Ångström exponent from samples collected at Union Glacier ranged from 1 to 3. These values are considerably lower than those from the Antarctic Peninsula, which is compatible with the fact that the Ellsworth Mountains are less dusty than the archipelagos off the Antarctic Peninsula.

A 2.6 m depth snowpit (Fig. 3b, c) enabled us to test potential year-to-year changes in the BC deposition on Union Glacier. BC concentration appears to have been decreasing in recent years from a maximum of about 3 ng/g during the 2013–2014 season (Fig. 3d), when the Chilean Union Glacier Camp became operational. The interannual changes in the absorption Ångström exponent (Fig. 3e) roughly followed (although in opposite direction) the changes in the BC concentrations (Fig. 3d). This suggests that changes in BC deposition drive year-to-year changes in light absorption while dust accumulation remains roughly constant. Since the Ångström exponent is an indicator of the relative contribution of dust to light absorption, it is plausible to attribute the peak during 2011–2012 to low BC deposition during that season (Fig. 3e).

**Other light-absorbing impurities**. The absorbing particles on the filters were not always a combination of BC and dust only,

**Table 1 Sampling sites.**

| Site | Latitude (°) | Longitude (°) | Season | Observations |
|---|---|---|---|---|
| *South Shetland Islands* | | | | |
| King George Island | | | | King George Island (95 km long and 25 km wide, with an area of 1150 km²) is the largest of the South Shetland Islands, lying 120 km off the coast of Antarctica in the Southern Ocean. Research stations operated by Argentina, Brazil, Chile, China, South Korea, Peru, Poland, Russia, and Uruguay are located on the island, where Chile operates one of the busiest Antarctic airfields. Of the visitors who travelled with IAATO members to the region in the season 2019/20, about 6% took at least one flight in or out of King George Island, combined with a Peninsula cruise. We sampled at 8 sites on the island including at two of the three major bays (Maxwell Bay and Admiralty Bay Arctowski) as well as on Collins Glacier, which covers about 75% of the island. |
| Arctowski Ice Field | −62.1203 | −58.6453 | 19/20 | |
| Collins Glacier 1 | −62.1673 | −58.8547 | 17/18 | |
| Collins Glacier (Dome) | −62.1686 | −58.8753 | 19/20 | |
| Collins Glacier 2 | −62.1698 | −58.8586 | 16/17 | |
| Collins Glacier (Artigas) | −62.1833 | −58.8844 | 19/20 | |
| Maxwell Bay (LARC) | −62.2028 | −58.9608 | 16/17 | |
| Maxwell Bay (Ardley) | −62.2167 | −58.9333 | 16/17 | |
| Maxwell Bay (South) | −62.2238 | −58.9635 | 17/18 | |
| Robert Island | | | | Robert Island (132 km²) is situated between Nelson Island and Greenwich Island. We sampled in the Coppermine Peninsula, located on the southwest coast of the island. |
| Coppermine Peninsula | −62.3786 | −59.7119 | 16/17 | |
| Greenwich Island | | | | Greenwich Island (142.7 km²) lies between Robert Island and Livingston Island. The Chilean Prat base and an Ecuadorian Maldonado base are situated on the northeast and north coast of the island, respectively. We sampled about 1 km west of the Prat station. |
| Prat Station | −62.4798 | −59.6434 | 18/19 19/20 | |
| Half Moon | | | | Half Moon Island (1.7 km²) lies 1.35 km north of Livingston Island in the South Shetland Islands. An Argentinian naval station is operational occasionally during the summer, but it is closed during the winter. |
| Half Moon Bay | −62.5956 | −59.9122 | 18/19 | |
| Livingston Island | | | | Livingston Island (798 km²; 73 km long and 36 km wide) has research stations on the northwest coast of Hurd Peninsula, operated in summer by Spain and Bulgaria. We sampled at 2 sites at Walker Bay near Hannah Point on the south coast of the island. Most of the extensive Byers Peninsula (about 60 km²), forming the west extremity of Livingston Island, is ice-free. |
| Walker Bay 1 | −62.6364 | −60.6008 | 19/20 | |
| Walker Bay 2 | −62.6362 | −60.6010 | 19/20 | |
| Deception Island | | | | Deception Island is the caldera of an active volcano. The island has a scientific outpost with Argentine and Spanish research stations, and has become a popular tourist destination. We sampled at Mount Pond near the Whaler Bay on the southwest coast of the island and at a site nearby the Spanish Gabriel de Castilla Station. |
| Mount Pond | −62.9656 | −60.5522 | 19/20 | |
| G. de Castilla Station | −62.9768 | −60.6691 | 18/19 | |
| *Palmer Archipelago* | | | | |
| Trinity Island | | | | Trinity or Trinidad Island (240 km²; 24 km long and 10 km wide) is in the northern part of the Palmer Archipelago. We sampled at two sites close to the southernmost point of the island (the Skottsberg Point), which marks the west side of Mikkelsen Harbor. |
| Mikkelsen Harbor 1 | −63.8967 | −60.8025 | 16/17 | |
| Mikkelsen Harbor 2 | −63.9133 | −60.8150 | 18/19 19/20 | |

**Table 1 (continued)**

| Site | | Latitude (°) | Longitude (°) | Season | Observations |
|---|---|---|---|---|---|
| Doumer Island | Yelcho Station 1 | −64.8761 | −63.5783 | 19/20 | Doumer Island (30 km²) lies between the south portions of Wiencke Island and Anvers Island (where the U.S. Palmer Station lies). We sampled at three sites close to the southernmost point of the island, nearby the Chilean Yelcho station and about 25 km east of Palmer Station. |
| | Yelcho Station 2 | −64.8775 | −63.5831 | 19/20 | |
| | Yelcho Station 3 | −64.8779 | −63.5824 | 19/20 | |
| *Antarctic Peninsula* | | | | | |
| Trinity Peninsula | O'Higgins Station | −63.3225 | −57.8970 | 18/19 | Trinity Peninsula is the northernmost part of the Antarctic Peninsula. We sampled nearby the Chilean O'Higgins Station, located about 30 km southwest of Prime Head, the northernmost point of the Antarctic Peninsula. Hope Bay (Bahía Esperanza; 5 km long and 3 km wide) indents the northern tip of the Antarctic Peninsula. It is the site of the Argentinian settlement Esperanza Base. We sampled about 2 km southeast of Esperanza Base. |
| | Hope Bay | −63.4078 | −56.9911 | 19/20 | |
| Charlotte Bay | | −64.5001 | −61.7661 | 18/19 | Charlotte Bay is a bay on the west coast of the Antarctic Peninsula indenting the west coast of Graham Land in a southeast direction. We sampled nearby Portal Point, where a British hut lies. |
| Cuverville Island | Northern shore | −64.6803 | −62.6194 | 16/17 | Cuverville (3.6 km²) is a rocky island lying in Errera Channel between Arctowski Peninsula and the northern part of Rongé Island. We sampled on the north coast of the island. |
| Petermann Island | Groussac Refuge | −65.1758 | −64.1361 | 16/17 | Petermann (2 km²) is a small, low and rounded island lying off the northwest coast of Kiev Peninsula, a short distance south of Booth Island and the Lemaire Channel. It is a popular tourist destination. We sampled nearby the Argentinian Groussac refuge. |
| | | | | 18/19 | |
| Detaille Island | Station W | −66.8687 | −66.7833 | 16/17 | Detaille is a small island off the northern end of the Arrowsmith Peninsula. We sampled nearby the unoccupied "Base W" of the British Antarctic Survey. |
| *Ellsworth Mountains* | | | | | |
| Union Glacier | Union Glacier 1 | −79.7669 | −82.9144 | 16/17 | Union Glacier is a large glacier that receives the flow of several tributaries and drains through the middle of the Heritage Range, Ellsworth Mountains. The glacier is the site of a blue-ice runway and two seasonally occupied camps (a Chilean camp and another operated by a private company that provides private expedition support and tours). In addition to the blue-ice runway, both camps operate ski-equipped airplanes. We sampled at two sites northwest of the Chilean camp, about 6 km east of the Blue-Ice Runway but less than 1 km away from designated landing sites for ski-equipped airplanes. |
| | Union Glacier 2 | −79.7625 | −82.9603 | 18/19 | |

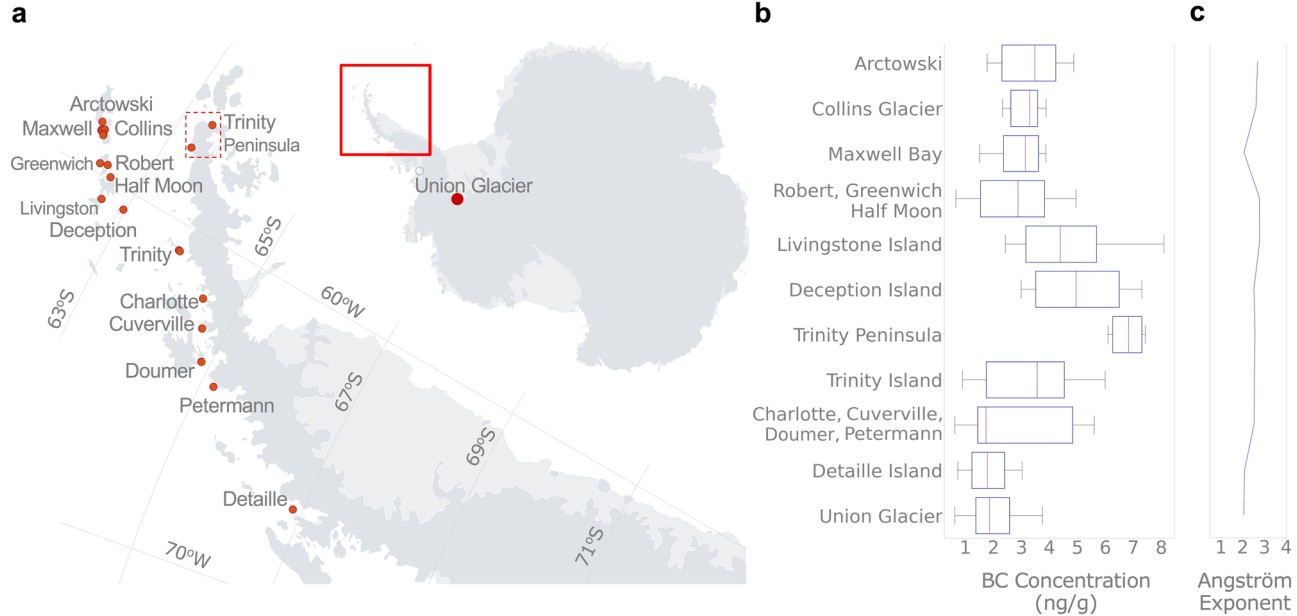

**Fig. 2 Black Carbon (BC) concentration around research stations and popular tourism destinations is considerable higher than elsewhere in Antarctica. a** Red dots represent the snow sampling sites. We sampled at 15 sites on the South Shetland Islands (where 16 research stations are located), five sites in the Palmer Archipelago, six sites on or near the coast of the Antarctic Peninsula, and at two deep-field sites on Union Glacier (Ellsworth Mountains); see Table 1 for details. **b** Boxplots of the BC concentrations in samples from different locations. In each box, the central mark (red stripe) indicates the median, and the edges indicate the 25th and 75th percentiles. The whiskers extend to the maximum and minimum data excluding outliers. **c** Mean of the absorption Ångström exponent of light-absorbing impurities in samples from different locations. Locations, given in order of latitude, in plots **b** and **c**, may combine samples from nearby sites. Measurements in plots **b** and **c** were subjected to the statistical significance tests whose results are presented in the Supplementary Information. Plots were generated by using Python's Matplotlib Library[58].

especially on the Antarctic Peninsula and associated archipelagos where snow algae are abundant on or near the coast[38]. Since algae also absorb light, especially in the visible range[39], we avoided sampling on algal fields (Supplementary Fig. 3a, b). Nevertheless, the distinctive presence of algal pigment appeared on the filters (Supplementary Fig. 3c) corresponding to some samples taken on Petermann Island, Robert Island, King George Island, and Doumer Island. The presence of algal biomass in snow from areas where it was not visible "to the naked eye" suggests that snow algae may be more abundant in Antarctica than prior estimates based on remote sensing[38]. It is apparent that the particulate absorption for the filters that exhibited vivid shades of red was strongly influenced by algal biomass. We used these samples for measuring the absorption optical properties of snow algae; we confirmed that light absorption attributable to algal biomass decreases rapidly with wavelength and plays no significant role in the absorption at wavelengths longer than 700 nm (which agrees with recent albedo measurements on algal fields[39]). The absorption Ångström exponent corresponding to samples with apparent algae presence ranged from 4 to 6, which is generally above the values of the absorption Ångström exponent corresponding to samples without apparent algae presence (Supplementary Fig. 4).

**Albedo reduction and snowmelt**. It is likely that local emissions account for most of the BC content in samples collected around research facilities and popular shore tourist-landing sites. BC concentration measured on the Antarctic Peninsula and associated archipelagos generally ranged from 2 to 4 ng/g, roughly corresponding to the 25th and 75th percentiles of samples collected in that region (Supplementary Table 3). These BC concentrations significantly exceed background levels (~1 ng/g, Supplementary Table 4) and are hardly attributable to intercontinental transport. Although massive wildfires in Patagonia can enhance BC depositions on the Antarctic Peninsula (last

occurring in 2015[40]), no major wildfire occurred in Patagonia over the period 2016–2020 (when our sampling campaigns occurred).

Although albedo reductions associated with the BC content in our snow samples are relatively low, any change in the extremely high albedo of Antarctic snow (Fig. 4a and Supplementary Fig. 5) affects the local radiative balance. In order to estimate albedo reductions attributable to local emissions, we adopted the parameterization proposed by Dang et al.[41] that provides, as a function of the cloud fraction (*CF*), estimates of the albedo reduction ($\Delta A$) for a range of BC mass mixing ratios and snow grain radii (*r*). Albedo reductions are generally favored by larger snow grains[41] and by clouds[4]. Although the snow of grain radii may exhibit a considerable dispersion (from 200 to 600 μm), the cloud fraction is generally close to 1 on the Antarctic Peninsula and associated archipelagos, one of the cloudiest regions on Earth.

According to the formulation by Dang et al.[41], the change in the BC concentrations attributable to local emissions on the Antarctic Peninsula and associated archipelagos can lead to albedo reductions ranging from 0.001 to 0.004 at BC-impacted sites. The lower limit (0.001) corresponds approximately to the albedo reduction expected if the BC concentration in snow of grain radii $r = 200$ μm changes from 1 ng/g (the background level) to 2 ng/g due to local emissions (Supplementary Fig. 6). The upper limit (0.004) corresponds approximately to the albedo reduction expected if the BC concentration in the snow of grain radii $r = 600$ μm changes from 1 ng/g (the background level) to 4 ng/g due to local emissions (Supplementary Fig. 6). Since the summer shortwave solar irradiance is on average about 250 W/m² on the Antarctic Peninsula and associated archipelagos (Fig. 4b), albedo reductions over the range 0.001–0.004 would lead to a positive local forcing of 0.25–1.00 W/m², which is comparable to the perturbation caused by the accumulation of some greenhouse gases in the atmosphere in recent decades.

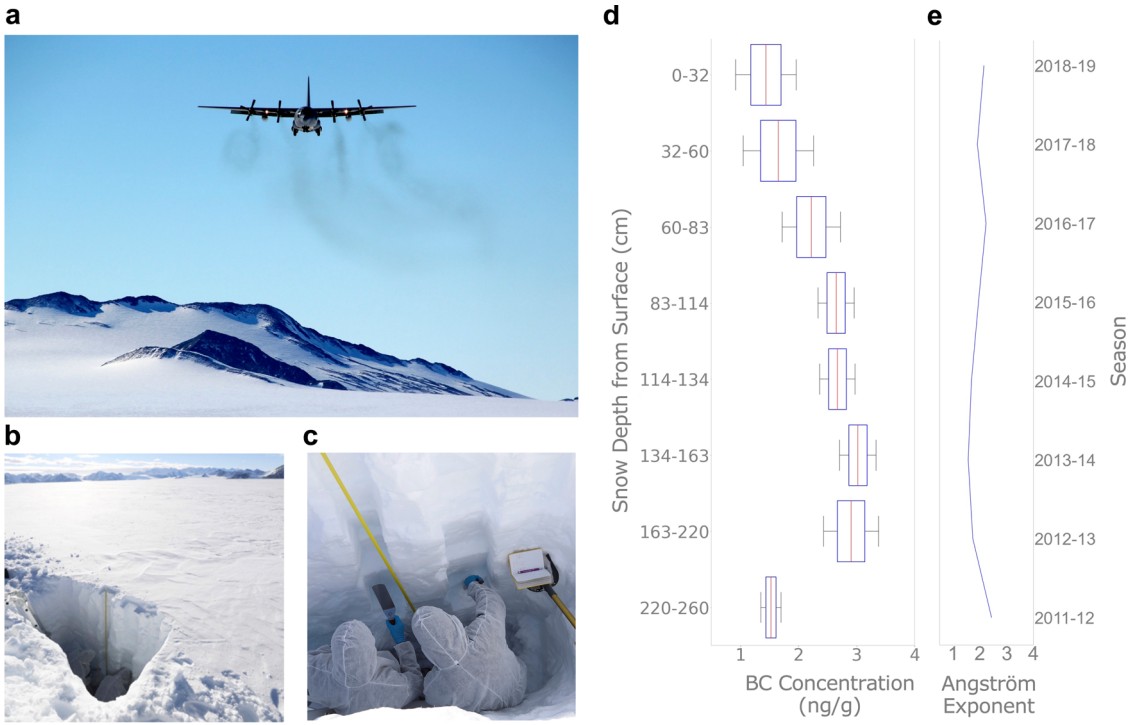

**Fig. 3 Black Carbon (BC) concentration is above background levels even around deep-field tourist destinations. a** C-130 aircraft about to land on the blue-ice runway at Union Glacier (79°S, Ellsworth Mountains), an increasingly popular deep-field destination (a private four-engine turbofan Ilyushin Il-76 landed on the runway 30 times in the 2019–2020 season only[34]). **b, c** Snowpit for snow sampling (6 km east of the blue-ice runway and about 1 km west of the designed landing sites for ski-equipped airplanes). **d** BC concentrations measured at different snow layers. BC concentration peaked at about 3 ng/g during the 2013–2014 season. Although private activities began years earlier, the Chilean Union Glacier Camp became operational during the 2013–2014 season. In each box, the central mark (red stripe) indicates the median, and the edges indicate the 25th and 75th percentiles. The whiskers extend to the maximum and minimum data excluding outliers. **e** Mean of the absorption Ångström exponent of light-absorbing impurities in snow from different layers. Photographs taken by the authors. Plots were generated by using Python's Matplotlib Library[58].

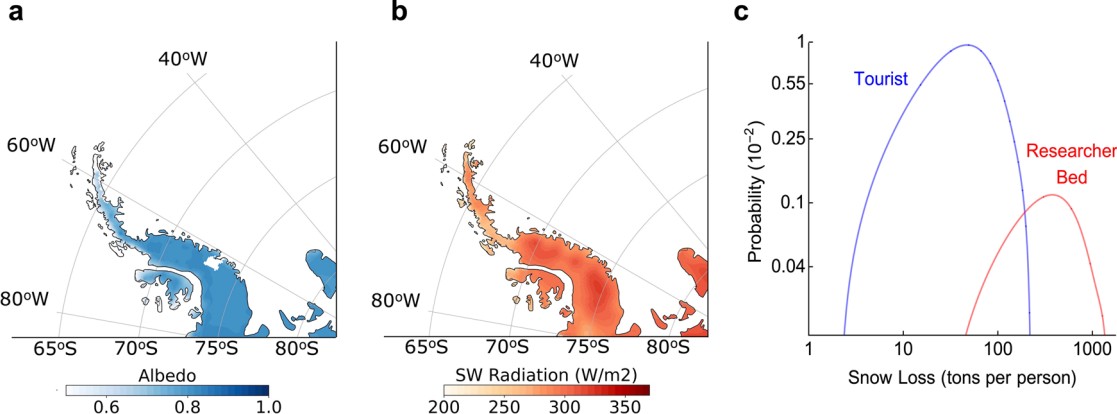

**Fig. 4 Black Carbon (BC) from local activities in Antarctica darkens the snow and makes it melt sooner. a** Broadband shortwave (SW) albedo averaged for December, January and February (DJF) days over the period 2004–2020. Data from MERRA-2[54] were used. **b** Downwelling shortwave (SW) all-sky irradiance averaged for DJF days over the period 1981–2019. The all-sky DJF SW irradiance is about 250 W/m². Data from the ERA5 Atmospheric Reanalysis[46] were used. **c** Probability density function of the snow that melts sooner due to the BC footprint of a researcher (red line) and a tourist (blue line). These probabilities are based on the simulations shown in Supplementary Figs. 7-8. In the case of the snow that melts sooner due to the BC footprint attributable to a researcher, we assumed that the area impacted by the presence of 11 research stations (with a total of 700 beds[33]) on King George Island ranges from 6 to 48 km². In the case of the snow that melts sooner due to the BC footprint attributable to a tourist, we assumed that the area impacted by BC emissions related to tourism ranges from 100 to 500 km²; we also considered that, on average, 53,000 tourists visited Antarctica annually from season 2016–2017 to the season 2019–2020[36]. Plots were generated by using Python's Matplotlib Library[58].

The positive forcing attributable to both research and tourism in Antarctica is accelerating snowmelt in BC-impacted areas (i.e., the snow-covered area where BC deposition occurred). We estimate that seasonal snow that melts sooner every summer at BC-impacted sites due to the associated albedo reduction ranges from 5 to 23 kg/m² (see Methods). This is equivalent to a snowpack shrinking of up to 23 mm w.e. The total mass loss attributable to the albedo reduction depends on how large the

BC-impacted area is. The latter is ultimately determined by how representative our snow samples are. Although limited spatial and temporal sampling could compromise our estimates, we can provide lower limits of snow mass loss assuming plausible values for the BC-impacted areas.

On King George Island, the largest of the South Shetland Islands, the snow that melts sooner every summer due to the BC footprint (largely attributable to research stations) is estimated to be 0.4 ± 0.2 Mt. We arrived at this figure by assuming that the BC concentration measured at each site is representative of a surrounding surface that ranged from 1 to 6 km². We sampled at 8 sites on King George Island such that we indirectly assumed that the BC-impacted area on King George Island was at most 48 km² (8 × 6), which is equivalent to 4% of the island's surface. This is a conservative estimation since King George is likely one of the most impacted islands in Antarctica. Research stations operated by Argentina, Brazil, Chile, China, South Korea, Peru, Poland, Russia, and Uruguay are located on the island, where Chile operates one of the busiest Antarctic airfields. According to COMNAP[33], there are eleven research stations on King George Island with a total of 700 beds, which means that the BC attributable to bed/researcher on the island accelerates melting of 0.6 ± 0.3 kt of snow every season (Fig. 4c).

Moreover, we estimate that the snow that melts sooner every summer due to the BC footprint of tourism on the Antarctica Peninsula and associated archipelagos is 4.4 ± 2.3 Mt. We arrived at this value by assuming that the area impacted by tourism ranges from 100 to 500 km² (note that 500 km² is around 0.1% of the surface of the Antarctic Peninsula and associated archipelagos). An average of 53,000 visitors travelled to Antarctica with IAATO operators from the season 2016–2017 to the season 2019–2020, which means that the BC attributable to an average visitor may accelerate summer melting of 83 ± 43 tons of snow (Fig. 4c).

## Discussion

Our systematic efforts to quantify the BC impact of local activities in Antarctica have revealed the burdens of BC deposition attributable to human activities (research activities and tourism) in the most visited region of the continent. Our surveys have shown that BC content in snow surrounding research facilities and popular shore tourist-landing sites (2 to 4 ng/g), while well below BC concentrations typical of other remote regions, is considerably above background levels measured elsewhere in the continent (~1 ng/g).

The radiative forcing owing to measured BC concentration is accelerating snowmelt and shrinking the snowpack of BC-impacted areas of the Antarctic Peninsula and associated archipelagos by up to 23 mm w.e. every summer. Burdens of BC deposition varied geographically, but we estimate that premature snowmelt due to the BC footprint of tourism is on the order of dozens to hundreds of tons per visitor (Fig. 4c). The intensive use of fuel-powered equipment at scientific stations makes the average snow losses attributable to each researcher at least one order of magnitude higher (Fig. 4c).

The BC footprint of research activities has likely increased as the construction of new facilities has risen in Antarctica. In recent years, there have been many new infrastructure projects on the Peninsula and associated archipelagos. For example, Brazil reconstructed its research station doubling its size, while the United Kingdom expanded the footprint of its Rothera station. Prior research has shown that buildings and infrastructure associated with science facilities displace wildlife and vegetation. Here, we have shown that research activities—which often include the use of helicopters, watercraft, airplanes, diesel generators, and all-terrain vehicles—are considerably darkening snow-covered areas next to research stations.

The BC footprint of tourism is likely driven by cruise emissions. A typical cruise includes several shore landings at the sites where we sampled. Cruise routes operated by IAATO often include exploring the volcanic islands of the South Shetland Islands and the Palmer Archipelago as well as visiting small islands off the Antarctic Peninsula such as Cuverville (64°S), Petermann (65°S) and Detaille (67°S). IAATO has been gaining an average of two to five operators every year and now includes more than 50 operators whose fleet of 54 vessels (including 6 large cruise ships) conducted a total of 378 departures in the 2019–2020 season[36].

The BC footprint per tourist was likely larger a decade ago, before the International Maritime Organization (IMO) banned the use of heavy fuel. Moreover, following IAATO advocacy for safe and environmentally responsible travel, some positive steps have been taken. Ships in Antarctica generally use less-polluting marine diesel, and some vessels are supplementing fuel with battery power. Nevertheless, our results show that more remains to be done to reduce the impacts of tourism and ships in Antarctica; otherwise, the burdens of BC deposition attributable to tourism will likely increase as human presence in Antarctica increases further.

Mechanisms to mitigate BC impacts are needed. IAATO should limit tourist activity and continue to push for a faster transition to clean fuel and hybrid or electric ships, while COMNAP should encourage National Antarctic Programs to limit the size and footprints of their research sites in Antarctica. Widespread adoption of energy efficiency standards and renewable power plants are urgently needed to limit the BC footprint of research facilities in Antarctica.

## Methods

**Sample collection and filtration**. Surveys were conducted during four consecutive seasons, from 2016–2017 to 2019–2020. We sampled at a total of 28 sites across a transect of about 2000 km from the northern tip of Antarctica (King George Island, 62°S) to the southern Ellsworth Mountains (Union Glacier Camp, 79°S); see Table 1 for further details.

Sampling was generally carried out in late summer, at the end of the melt season, mostly at coastal sites where the snow is often thin and patchy. Union Glacier was an exception; at this deep-field location, sampling occurred in early December. BC concentration in snow samples collected in late summer are expected to be the highest of the season since particles collect at the surface as the snow melts during summer;[42] BC normally concentrates at the surface in melting snow and only 10–30% of the BC is removed with meltwater[43]. We focused on summer because BC deposition in seasons other than summer is not consequential in terms of snowmelt or radiative balance.

We collected duplicates from the snow-covered surfaces, horizontally separated by around 1 m. The difference between these measurements was generally lower than 1 ng/g but we did occasionally find differences slightly higher than 1 ng/g. These differences may have resulted from actual variations in the BC content as soot content in Antarctica may vary considerably on horizontal scales of a few meters due to frequent blizzards[18]. At four sites on the Antarctica Peninsula, where the snow was abundant enough, snow was sampled vertically at intervals of 10–20 cm depth to study the vertical distribution of BC in snowpits that captured the entire snowpack (up to 1 m depth from the surface). In some cases, the surface layer was cleaner than the subsurface layers, which is likely due to snowdrift or recent precipitation. However, BC concentrations were found to be generally comparable within a profile (of the same order). At Union Glacier Camp, a deeper snowpit (up to 2.6 m depth) enabled us to test potential year-to-year changes in the BC concentration. The masses of samples ranged from 1 to 2 kg each.

As our aim was to assess the influence of light-absorbing impurities on the albedo of representative areas, sampling sites were chosen to always be hundreds of meters (even kilometers when possible) away from apparent pollution sources such as research stations, tourist shore landings, roads and airfields. Moreover, particular attention was paid to preventing contamination of the samples. A stainless-steel spatula was used for collection and samples were placed into plastic bags, which were in turn packed in Whirlpack bags. Styrofoam coolers were used for transportation to the laboratory. In the laboratory deionized water was used to wash tools and containers before filtering, and latex gloves and lab coats were worn during filtering. Snow samples were transferred to a glass beaker and snowmelt was subsequently vacuum-filtered, leaving the insoluble material on the filter. Stainless-

steel funnels and 0.4 µm Nucleopore filters were used. Filters were subsequently placed in sterile petri dishes. Our field blank analyses showed that the effect of contamination during the sampling, storage and transportation was on the order of tenths of ng/g.

**BC concentration and Ångström exponent.** We applied the Meltwater Filtration (MF) Technique, initially proposed by Clarke and Noone[7], that has been broadly applied for measuring BC concentrations in snow samples from around the world (from the Arctic[4–7], North America[8,9], Northern China[10], the Andes[13], and the South Pole[18]). The MF technique's measures of absorption are closely related to the sunlight absorption in the snowpack[44].

Following the procedure described by Grenfell et al.[44], filters were analyzed for light transmission. We measured over the range 340–750 nm at a 5 nm resolution using a spectrophotometer. These measurements were used to derive the maximum possible BC concentration ($C_{MAX}$), the best estimate of the BC concentration ($C_{EST}$), and the spectral absorption properties of light-absorbing particles in samples (i.e., the Ångström exponent).

Spectrophotometric measurements allowed us to compute the absorbance ($A$) taken as equal to the inverse of the logarithm of the transmittance ($T$) of the spectral radiant power through the filters:

$$A = -\log T. \tag{1}$$

The absorbance at each wavelength over the range 340–750 nm was then converted to a BC-equivalent loading ($L$) by using wavelength-dependent calibration curves. These calibration curves were previously obtained from measurements of the light transmission through filters loaded with known amounts of synthetic Monarch-71 soot, often taken as a proxy for BC in nature. A comparison of the light attenuation by a sample filter to the light attenuation by the synthetic standards yields the BC-equivalent loading of the sample filter. The wavelength-dependent BC-equivalent loading ($L$) was then converted to a BC-equivalent concentration ($C$) by accounting for the filter-exposed area ($S$) and the mass of snow meltwater from each sample ($V$):

$$C = L.\frac{S}{V}. \tag{2}$$

The BC-equivalent concentration generally varies with wavelength because the sample filter usually contains non-BC light-absorbing particles as well as BC particles, whose spectral responses differ. However, absorption by non-BC constituents decreases more rapidly with wavelength than does absorption by BC. Accordingly, both dust[45] and algae[39] play no significant role in the absorption at wavelengths longer than 700 nm and concentrations computed by Eq. (2) exhibit few changes over the range 700–750 nm.

Since most of the light absorption observed over the range 700–750 nm is attributable to BC, here we estimated the maximum possible BC concentration ($C_{MAX}$) for each sample by averaging the wavelength-dependent concentration computed by Eq. (2) over the range 700–750 nm.

The spectral absorption properties of light-absorbing impurities in samples can be characterized by fitting the absorption Ångström law to the wavelength-dependent absorption optical depth (AOD):

$$AOD(\lambda) = \beta.\lambda^{-\alpha}, \tag{3}$$

where $\lambda$ is the wavelength in µm, and $\alpha$ and $\beta$ are referred to as Ångström parameters. At each wavelength over the range 340–750 nm, the AOD is the mass absorption coefficient (MAC) of Monarch-71 soot multiplied by the BC-equivalent loading ($L$). MAC estimates of fullerene are 9.8 m²/g at 400 nm, 7.2 m²/g at 550 nm, and 5.9 m²/g at 650 nm[8]. According to Eq. (3), the absorption Ångström exponent ($\alpha$) was calculated for each sample as a linear fit over the range 420–620 nm to the plot AOD-$\lambda$ in log-log space. Prior efforts[6,43] have shown that the absorption Ångström exponent of pure BC ($\alpha_{BC}$) is 1.1. The deviation of the absorption Ångström exponent from 1.1 is, therefore, an indication of the amount of absorption by non-BC particles in the snow.

Calculation of the best estimate of the BC concentration ($C_{EST}$) in samples requires attribution of total measured absorption to BC and non-BC constituents. In order to separate the absorption contributions, we followed Grenfell et al.[43] expanding the absorption Angström exponent ($\alpha$) around the midpoint ($\lambda_o$= 520 nm) of the range 340–700 nm:

$$\alpha = z.\alpha_{BC} + (1 - z).\alpha_{NBC}, \tag{4}$$

where $z$ stands for the fraction of absorption due to BC in samples, $\alpha_{BC} = 1.1$, and $\alpha_{NBC}$ is the Ångström exponent of non-BC light-absorbing particles (i.e., crustal dust) that, according to prior research[10], we took to be equal to 4. Rearranging Eq. (4) yields $z$, which can, in turn, be used to compute $AOD_{BC}$ at $\lambda_o$= 520 nm:

$$AOD_{BC}(\lambda_o) = z.AOD(\lambda_o), \tag{5}$$

where $AOD(\lambda_o)$ is the MAC of Monarch-71 soot multiplied by the BC-equivalent loading ($L$) at $\lambda_o$= 520 nm. $AOD_{BC}(\lambda)$ is related to the Ångström exponent of pure BC ($\alpha_{BC}$) by the absorption Ångström law:

$$AOD_{BC}(\lambda) = \beta.\lambda^{-\alpha_{BC}}, \tag{6}$$

where $\beta$ can be written as a function of the $AOD(\lambda_o)$ by rearranging the Ångström

law:

$$\beta = AOD_{BC}(\lambda_o).\lambda_o{}^{\alpha_{BC}}. \tag{7}$$

The wavelength-dependent absorption optical depth corresponding to the BC in samples was estimated by combining Eqs. (6) and (7):

$$AOD_{BC}(\lambda) = AOD_{BC}(\lambda_o).\left(\frac{\lambda}{\lambda_o}\right)^{-\alpha_{BC}}. \tag{8}$$

Finally, since the absorption over the range 700–750 nm is attributable to BC, we computed the best estimate of the BC concentration ($C_{EST}$) in each sample according to

$$C_{EST} = C_{MAX}.\left(\frac{AOD_{BC}(700:750\,nm)}{AOD(700:750\,nm)}\right), \tag{9}$$

where

$AOD_{BC}(700:750\,nm)$ is the average of $AOD_{BC}$ values computed by Eq. (8) over the range 700–750 nm, and

$AOD(700:750\,nm)$ is the average of the AOD values computed over the range 700–750 nm.

Values of the Absorption Ångström exponent ($\alpha$) and BC concentrations ($C_{EST}$) were used to build up Fig. 2. Details on the statistical significance of the data in Fig. 2 are shown in the Supplementary Information (Supplementary Tables 1-2).

**Albedo reduction, radiative forcing, and snowmelt.** According to Dang et al.[4], the all-sky broadband albedo reduction ($\triangle A$) due to light-absorbing impurities in snow can be estimated as

$$\triangle A = CF.\triangle A_{cloudy} + (1 - CF).\triangle A_{clear}, \tag{10}$$

where $CF$ is the cloud fraction and $\triangle A_{cloudy}$ and $\triangle A_{clear}$ are the albedo reductions under overcast and cloudless conditions, respectively. Dang et al.[41] provide estimates of both $\triangle A_{cloudy}$ and $\triangle A_{clear}$ for a range of BC mass mixing ratios and snow grain radii (Supplementary Fig. 6). Albedo reductions were not calculated for every site, but rather for regional averages.

The radiative forcing for summer months (DJF) was calculated as the albedo reduction times the all-sky shortwave irradiance ($I$). For both $I$ and $CF$, we used estimates for summer months (DJF) from the ERA5 Atmospheric Reanalysis[46] over the period 1981–2019, averaged over the region of interest. For the Antarctic Peninsula and associated archipelagos, one of the cloudiest regions on Earth, $I$ is typically about 250 W/m² (Fig. 4b) while $CF$ approaches 1 (Supplementary Fig. 7), such that and Eq. (10) reduces to:

$$\triangle A = \triangle A_{cloudy}. \tag{11}$$

The snow ($W$) that melts sooner per unit of the surface due to the albedo reduction ($\triangle A$) during summer in that region was estimated according to

$$W = \frac{\triangle A.I.d}{E}, \tag{12}$$

where $d$ is a period of interest (that we took equal to the 90 DJF days) and $E$ is the energy needed to melt 1 kg of snow (which we took to be equal to the latent heat of fusion, 334 kJ/kg[47]).

Finally, the mass of the snow ($W_t$) that melted sooner due to the albedo reduction ($\triangle A$) associated to local emissions was estimated as

$$W_t = \frac{\triangle A.I.d}{E}.D, \tag{13}$$

where $D$ is the BC-impacted area (i.e., the snow-covered area where BC deposition occurred).

**Snowmelt estimation for BC-impacted sites on the Antarctic Peninsula.** We applied Eq. (12) and Eq. (13) to estimate the snowmelt attributable to the local BC emission, which first required estimating the albedo reduction ($\triangle A$) associated with the BC content detected in samples collected around research stations and popular shore tourist-landing sites on the Antarctic Peninsula and associated archipelagos.

The albedo reduction ($\triangle A$) depends on the cloud fraction ($CF$), the snow grain radii ($r$), and the BC concentration ($C_{EST}$). While the cloud fraction over the Antarctic Peninsula and associated archipelagos is generally close to 1, the snow grain radii are strongly influenced by the temperature and may exhibit a considerable dispersion (from 200 to 600 µm). So does the BC concentration; our results show that the BC content detected in samples collected around research facilities and tourist-landing sites generally ranged from 2 to 4 ng/g (roughly corresponding to the 25th and 75th percentiles of samples collected on the Antarctic Peninsula and associated archipelagos; Supplementary Table 3).

According to the formulation by Dang et al.[41] and considering the predominantly overcast conditions in our region of interest, the change in the BC concentrations attributable to local emissions can lead to albedo reductions over the range 0.001–0.004 at BC-impacted sites. The lower limit (0.001) corresponds approximately to the albedo reduction expected if the BC concentration in the snow of grain radii $r = 200$ µm changes from 1 ng/g (the background level) to 2 ng/g due to local emissions (see Supplementary Fig. 6a). The upper limit (0.004)

corresponds approximately to the albedo reduction expected if the BC concentration in the snow of grain radii $r = 600\ \mu m$ changes from 1 ng/g (the background level) to 4 ng/g due to local emissions (see Supplementary Fig. 6a).

Since the summer shortwave solar irradiance is on average about 250 W/m$^2$ in the region (Fig. 4b), albedo reductions over the range 0.001–0.004 would lead to a positive local forcing of 0.25–1.00 W/m$^2$, which will, in turn, make the local snowpack absorb on average an extra of 2–8 MJ/m2 over the 90-day summer period. The enormous amount of extra energy absorbed by the snowpack will melt not only the snow at the zero-degree isotherm but also at sites where the air temperature is slightly below 0 °C.

Equation (12) allowed to estimate the snow ($W$) that melts sooner per unit of the surface at BC-impacted sites on the Antarctic Peninsula and associated archipelagos. Considering again that the summer shortwave solar irradiance is on average about 250 W/m$^2$ in our region (Fig. 4b), we found that albedo reductions over the range 0.001–0.004 would lead to the snowmelt of 5–23 kg/m$^2$ (which is equivalent to a snowpack shrinking over the range 5–23 mm w.e.) over the 90-day summer period.

We applied Eq. (13) to estimate the mass of the snow ($W_t$) that melted sooner due to the albedo reduction ($\triangle A$) associated with local emissions. The total mass loss attributable to the albedo reduction depends on how large the BC-impacted area ($D$) is. The BC-impacted area can range from a few square kilometers to dozens of square kilometers around a BC source. In the case of King George Island, we assumed that the BC-impacted area ranges from 6 to 48 km$^2$ (about 0.7–4% of the island's surface). In the case of the area impacted by tourism, we assumed that it ranges from 100 to 500 km$^2$ (around 0.02–0.10% of the surface of the Antarctic Peninsula and associated archipelagos).

In order to account for the effect of the dispersion of both $\triangle A$ and $D$, we conducted a Monte Carlo simulation[48,49]. Accordingly, we recursively applied Eq. (13) by using large sets of previously generated values of $\triangle A$ and $D$. In the case of King George Island, we randomly generated values of $\triangle A$ and $D$ over the ranges 0.001–0.004 and 8–48 km$^2$, respectively (Supplementary Fig. 8). In the case of the area impacted by tourism, we randomly generated values of $\triangle A$ and $D$ over the ranges 0.001–0.004 and 100–500 km$^2$, respectively (Supplementary Fig. 9). Pairs of these randomly generated values were used as inputs in Eq. (13). The histograms of the results are shown in Supplementary Fig. 8c and Supplementary Fig. 9c. The mean and the standard deviations of the $W_t$ values in Supplementary Fig. 8c were used for estimating the snow that melts sooner every summer due to BC deposition from local sources on King George Island: 0.4 ± 0.2 Mt, which translates to about 0.6 ± 0.3 kt of snow per bed; there are 11 research stations with a total of 700 beds on King George Island[33]. Similarly, the mean and the standard deviations of the $W_t$ values in Supplementary Fig. 9c were used for estimating the snow that melts sooner every summer due to BC emissions associated with tourism on the Antarctic Peninsula and associated archipelagos: 4.4 ± 2.3 Mt, which translates to about 83 ± 43 tons per visitor; an average of 53,000 tourists visited the region from the season 2016–2017 to the season 2019–2020[36].

Finally, we used the histograms shown in Supplementary Fig. 8c and Supplementary Fig. 9c to infer the probability density functions (Fig. 4c) of the snow that melts sooner due to the BC footprint of a researcher bed and a tourist. Figure 4c shows that the BC footprint of tourism in Antarctica is on the order of dozens to hundreds of tons per visitor (blue line); the intensive use of fuel-powered equipment at scientific stations makes average snow losses attributable to each researcher in Antarctica at least one order of magnitude larger (red line).

**Back-trajectory analysis**. Back-trajectory analysis suggests that long-range transport (from Patagonia for example) has only a minor role as a source of absorbing aerosols in Antarctica. We applied air-parcel backward trajectories to selected sampling sites. We used the Hybrid Single-Particle Lagrangian Integrated Trajectory (HYSPLIT) model[50,51] fed with the Global Data Assimilation System (GDAS) archive (available in a resolution of three hours and a 1° latitude-longitude)[52]. For selected sites, we computed 72 h backward trajectories for DJF days over the period 2010–2020 (Supplementary Fig. 10). For the same sites, we separately computed 72 h backward trajectories for a 3-month period before sampling (Supplementary Fig. 11). These trajectories were divided into clusters following a cluster analysis based on the total spatial variance (TSV)[53]. The results, shown in the Supplement Information for 4 representative sampling sites, support the premise of this work: most of the BC in our samples was emitted locally.

**Reanalysis**. Broadband shortwave (SW) albedo data were obtained over the period 2004–2020 from Modern-Era Retrospective analysis for Research and Applications (MERRA-2)[54]. Downwelling shortwave (SW) all-sky irradiance and cloud fraction (CF) data were obtained over the period 1981–2019 from the ERA5 Atmospheric Reanalysis[46].

**Statistical significance tests**. Site grouping (Supplementary Fig. 2) was tested by using an Analysis of Variance (ANOVA)[55] and the Tukey's Honestly Significant Difference (HSD) Test[56]. The results of ANOVA (Supplementary Table 1) and the Tukey's HSD Test (Supplementary Table 2) confirm that, for example, measurements of the BC concentration and the Ångström exponent at sites on Union Glacier are significantly different from measurements at sites on the Antarctica

Peninsula and associated archipelagos (which was expected since the sites on Union Glacier are the only ones in the vast interior of the continent). We also used one-way t-tests confirming that BC concentrations are significantly different from 0 or 1 at each site grouping (Supplementary Table 4). All statistical analyses were conducted in JMP 15.2.0[57].

## Data availability

MERRA-2 data are available at https://disc.gsfc.nasa.gov/datasets?project=MERRA-2. ERA5 data are available at https://cds.climate.copernicus.eu/cdsapp#!/dataset/derived-near-surface-meteorological-variables?tab=overview. The data (BC content and Ångström Exponent) generated in this study are available at http://antarctica.cl/bc-measurements.

## Codes availability

The Python codes (version 3.6) used in this study are available at http://antarctica.cl/bc-code/

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

## Acknowledgements

The support of INACH (RT_69-20 & RT_70-18), ANID (ANILLO ACT210046, FON-DECYT 1191932, DFG190004, and REDES180158), CORFO (Preis 19BP-117358, 18BPE-93920, and 18BPCR-89100), Dicyt-USACH, and Antarctica XXI is gratefully acknowledged.

## Author contributions

R.R.C., S.F., A.D., J.C., G.S., P.M.R., S.N., and A.L.K.: wrote the main manuscript text. S.F., E.S., C.W., A.M., and F.F. prepared figures. R.R.C., S.F., E.S., F.F., V.A., J.A.A., P.L., P.W., M.S., J.M.C., J.J., J.D., A.L.K., and G.C. collected and analyzed samples. All authors reviewed the manuscript.

## Competing interests

The authors declare no competing interests.
