## [Peer Review File · Nature Communications]

Black Carbon Footprint of Human Presence in AntarcticaREVIEWER COMMENTS

Reviewer #1 (Remarks to the Author):

Quantifying how human activity in Antarctica contributes to BC concentration in the snow and ice and the resulting effect on albedo and melt is an important study. The methods used to collect snow samples and measure BC, and other light absorbing impurities, in the snow are scientifically sound. The authors clearly explain how BC is differentiated from dust and snow algae using known absorptive spectra and AAE values. Conclusions regarding the resulting effect on snow albedo and ice melt in the region is a stretch given the substantial uncertainties that are either not addressed or glossed over in this manuscript. As the analysis of additional melt due to BC boosts the significance of the study, the authors should provide a more thorough explanation of the uncertainties and allow the editor to determine if publication in Nature is warranted. If the editor determines the paper is not well suited to Nature, the authors may want to consider focusing more on the measurements, and less on quantifying additional melt, when submitting to another journal. More details on the statistical significance, measurement uncertainties, interannual variability, and local vs continental influence of light absorbing aerosol in Antarctic snow would make an interesting and important paper without trying to quantify albedo reduction and resulting melt based on layered assumptions and parameterizations.

The following specific issues need to be addressed.

1) An in-depth discussion of the statistical significance of Figure 2 is needed. The discussion of albedo reductions and snow melt hinge on these measurements. Uncertainties grow exponentially with assumptions made during the progression from BC concentration in snow to albedo change to melt rate. If measurement uncertainty is the same magnitude as concentration, any discussion of albedo reduction or melt rate is inappropriate.

- Does figure 2 represent the range of sample concentrations from each site (or site grouping) during all 4 seasons? It would be useful to see these box/whisker plots at each site for each season. If there were sufficient samples collected and analyzed during each season, interannual variability would either show persistence in the results or reveal if one year dominated the result. If there was one year that stands out, the analysis would also show whether only one or all sites were affected.

- Grouping the results from sites that are in proximity is appropriate. Results from sites like Peterman and Demaille should be separated.

- Were enough duplicate samples collected and analyzed such that there is an uncertainty estimate in the measurement method for each sample? Given the presence of dust and snow algae, is 1 to 2 ng/g of BC significantly different from zero? Is 2 to 4 ng/g significantly different than 1 ng/g of BC in snow as stated in lines 268 – 270?

2) Once the uncertainty in the measurements is quantified, the uncertainty of BC induced snow albedo reduction needs to be addressed.

- When converting an absorption measurement on a filter to equivalent BC mass, a mass absorption coefficient (MAC) value must be prescribed (lines 397 – 425). The authors use a value obtained from the Monarch-71 soot standard. There are uncertainties around the use of this value, although the authors indicate the calculated BC concentration is really equivalent to the concentration of Monarch-71 soot (line 374) that would cause the measured absorption (eqn 1, line 365) by the particle laden filter. If this is the case, can you verify that the BC used in the albedo reduction model (Dang et al, supplemental info) is consistent with the MAC of Monarch 71 soot? These need to be the same or the measured concentrations used to feed the model would be inaccurate. The other option is to propagate inherent uncertainty in using a prescribed MAC through the calculations and add them to measurement uncertainties mentioned above. These uncertainties are critical given the very low BC concentrations reported in this study.

- On lines 219-220, the authors claim that 1-3 ng/g of BC can reduce albedo by 0.001 – 0.004. A brief mention of the model used and the reference to Dang et al. should be included in the main text as well as supplementary information).

3) The relative impact albedo has on ice melt is most significant when ambient temperature is near the melting point. When ambient temperature is significantly colder or warmer, the tiny changes in albedo due to the small concentrations of BC found in this study would likely be insignificant. The authors should include a discussion of ambient temperature on the melt results calculations.

4) The authors unequivocally state on lines 229 and 239 that BC is accelerating snow melt. While

this might be true, they have not measured it or proven this to be the case. They need to make clear that they are hypothesizing snow is melting earlier due to BC concentrations.

5) While the authors cite other studies showing long-range transport plays a minor role as a source of light absorbing aerosols in Antarctica, back trajectories specific to their samples should be calculated. The Antarctic peninsula is not that far from Argentina and downwind from Australia. One bad dust storm or smoke event could deposit enough light absorbing aerosol on the snow in a season to impact the entire study, even if it is short-lived and rare. The authors claim local activity is the primary source of light absorbing aerosol over a wide area of Antarctica but have failed to back up this statement for their data specifically. If a seasonal analysis as recommended above were to reveal strong interannual variability at the sites, authors should look to see if continental influence could have played a role.

6) The Union Glacier site while furthest from the peninsula does not have the lowest BC concentrations as the authors claim on line 171. Figure 2 shows that statistically the lowest BC concentrations were found at the Peterman and Demaille sites, identified as popular tourist sites with no permanent settlements.

- Do any of the site measurements represent background Antarctica unaffected by local camps/activity?
- More details about the level of human activity at each site would help support the authors' claim that it is local activity and not continental influence that is the primary source of light absorbing aerosols.

In summary, the data are interesting and a more thorough analysis into the statistical significance, interannual variability in addition to inter-site variability, and an examination of potential sources would strengthen the findings considerably. In current form, the paper is not suitable for publication in Nature.

Reviewer #2 (Remarks to the Author):

This manuscript presents the black carbon and other light-absorbing particles measured across 15 snow-covered Antarctica sites around research stations and popular tourist destinations. The authors quantified the concentrations of light-absorbing particles and attributed their sources to research or tourist activities. The authors adopted a mature filter-based technique and sampling procedure that has been applied to snow/BC sampling globally. The manuscript reads well, the analyses are accurate, and the discussions are clear. The reviewer does not have any additional comments and recommends this paper for publication.

Suggestions of Reviewer 1

Comment of Reviewer 1

Quantifying how human activity in Antarctica contributes to BC concentration in the snow and ice and the resulting effect on albedo and melt is an important study. The methods used to collect snow samples and measure BC, and other light absorbing impurities, in the snow are scientifically sound.

We thank the reviewer for her/his helpful suggestions and constructive remarks. We agree with the reviewer on the urgent need for assessing the effects of Black Carbon (BC) deposition in Antarctica.

Comment of Reviewer 1

The authors clearly explain how BC is differentiated from dust and snow algae using known absorptive spectra and AAE values. Conclusions regarding the resulting effect on snow albedo and ice melt in the region is a stretch given the substantial uncertainties that are either not addressed or glossed over in this manuscript. As the analysis of additional melt due to BC boosts the significance of the study, the authors should provide a more thorough explanation of the uncertainties and allow the editor to determine if publication in Nature is warranted. If the editor determines the paper is not well suited to Nature, the authors may want to consider focusing more on the measurements, and less on quantifying additional melt, when submitting to another journal. More details on the statistical significance, measurement uncertainties, interannual variability, and local vs continental influence of light absorbing aerosol in Antarctic snow would make an interesting and important paper without trying to quantify albedo reduction and resulting melt based on layered assumptions and parameterizations.

This is a very good point and we thank the reviewer for raising it. We agree with the reviewer that our focus could have been on the measurements. After all, we report on the first comprehensive survey of the BC footprint of local anthropogenic activities in Antarctica; an unprecedented multi-national effort for measuring the BC concentration in Antarctic snow from 28 sites from the northern tip of Antarctica (62°S) to the southern Ellsworth Mountains (79°S).

However, we believe that our study makes a much stronger contribution by also assessing the so far largely ignored effect of local BC emissions on Antarctic snow melting. We are confident that the unique dataset that resulted from our campaigns provides enough information to evaluate the burdens of BC deposition on the Antarctic snow surrounding research stations and popular tourist destinations.

We agree of course with the reviewer that the thorough statistical analyses should be provided, and we have accordingly modified the attached version of our manuscript (see further details below). Also, we have highlighted the fact that, as pointed out by reviewer 2, one of the strengths of our manuscript is that we “adopted a mature filter-based technique and sampling procedure that has been applied to snow/BC sampling globally”. In fact, the MF technique has been previously applied for measuring BC content in snow sampled in Greenland (Doherty et al., 2010; Hegg et al., 2010), the Arctic (Dang et al., 2017; Clarke & Noone, 1985), Antarctica (Warren & Clarke, 1990), North America (Doherty et al., 2014; 2016), Northern China (Wang et al., 2013), and the Andes (Rowe et al., 2019).

Comment of Reviewer 1

1) An in-depth discussion of the statistical significance of Figure 2 is needed. The discussion of albedo reductions and snow melt hinge on these measurements. Uncertainties grow exponentially with assumptions made during the progression from BC concentration in snow to albedo change to melt rate. If measurement uncertainty is the same magnitude as concentration, any discussion of albedo reduction or melt rate is inappropriate.

• Were enough duplicate samples collected and analyzed such that there is an uncertainty estimate in the measurement method for each sample? Given the presence of dust and snow algae, is 1 to 2 ng/g of BC significantly different from zero? Is 2 to 4 ng/g significantly different than 1 ng/g of BC in snow as stated in lines 268 – 270?

This is another good point. We agree that any further discussion is pointless if the uncertainty associated with our measurements is as large as the measured values. Fortunately, this is not the case.

As extensively discussed by Grenfell et al., (2011), the uncertainty associated with BC measurements conducted by the Meltwater Filtration (MF) technique is below 10%, which in our case is equivalent to a few tenths of ng/g (0.2-0.4 ng/g). This is why BC measurements conducted by the MF technique even at extremely clean sites, such the Amundsen–Scott South Pole Station (Warren & Clarke, 1990), are considered reliable despite the fact they were generally lower than 1 ng/g.

We are also quite confident that the BC concentrations shown in Fig. 2 are significantly different from zero and significantly above 1 ng/g. Our field blank analyses showed that the effect of contamination during the sampling, storage and transportation was also on the order of tenths of ng/g. In addition to methodological checks, we also carried out statistical tests. For example, we conducted a one-way t-tests confirming our BC measurements are different from 0 or 1 ng/g (see additional details below).

At each site, we collected duplicates from the snow-covered surfaces, horizontally separated by about 1 m. The difference between these measurements was generally lower than 1 ng/g but we did occasionally find differences slightly higher than 1 ng/g. However, note that these differences likely resulted from actual variations in the BC content. As shown by Warren & Clarke (1990), soot content in Antarctica may vary considerably on horizontal scales of a few meters due to frequent blizzards. The difference between these measurements taken at different depths in the snowpit also occasionally exhibited differences larger than 1 ng/g, which suggests that variability of the BC concentration on horizontal and vertical scales largely exceeds the uncertainty associated with individual measurements of the BC concentration.

According to the suggestion of the reviewer, in the new version of the Supplementary Material, we have included the results of the statistical significance tests that we conducted on the measurements shown in Fig. 2. In particular, we have included:

- 1) New tables with the results of an Analysis of Variance (ANOVA) (Table S1) and a Tukey's Honestly Significant Difference Test (Table S2) that we used to test site grouping (see Fig. S2). These tests confirm that, for example, measurements at sites on Union Glacier are significantly different from measurements at sites on the Antarctica Peninsula and associated archipelagos (which was expected since the sites on Union Glacier are the only ones in the vast interior of the continent);
- 2) A new table with the minimum and maximum, 25% and 75% quartile, and median, reported for each of the site groupings (Table S3).
- 3) A new table with the BC mean and standard deviation of each site grouping (Table S4). The standard deviation is particularly relevant since it characterizes the variability observed in the BC concentration at each site grouping and is far more representative of the actual dispersion of the BC concentration (than the relatively small uncertainty associated with individual measurements).
- 4) The results of one-way t-tests confirming that BC concentrations are significantly different from 0 or 1 at each site grouping (Table S4).

Comment of Reviewer 1

• Does figure 2 represent the range of sample concentrations from each site (or site grouping) during all 4 seasons? It would be useful to see these box/whisker plots at each site for each season. If there were sufficient samples collected and analyzed during each season, interannual variability would either show persistence in

the results or reveal if one year dominated the result. If there was one year that stands out, the analysis would also show whether only one or all sites were affected.

- Grouping the results from sites that are in proximity is appropriate. Results from sites like Peterman and Detaille should be separated.

As indicated in the section “Methods”, since we were interested in the assessment of the BC deposition attributable to local emissions, sampling was carried out in summer only. We focused on summer because:

- 1) the vast majority of activities in Antarctica (research and tourism) occurs in summer; and
- 2) BC deposition in seasons other than summer is not consequential in terms of snowmelt (or radiative balance) due to the absent/weak solar radiation.

The interannual variability is another good point raised by the reviewer. As shown in Table 1, due to the huge logistic challenges of getting samples out of Antarctica, we were able to sample twice (in different years) at only 3 coastal sites: Prat Station, Mikkelsen Harbor 1, and Groussac Refuge. At these sites, the differences between BC concentrations measured in samples from different years were always lower than 1 ng/g. Of course, samples from two different years are insufficient to render meaningful information regarding the interannual variability. That is why we included in our study a site in the continental interior (Union Glacier).

A 2.6-m depth snowpit (Fig. 3) enabled us to test potential year-to-year changes in the BC deposition on Union Glacier. At this deep-field site, there is no considerable surface melting and the snow accumulates every year. Accordingly, by analyzing samples collected at different depths in the snowpit, we were able to study the interannual variability of the BC deposition from 2011 to 2019 at this site. This type of analyses cannot be conducted at coastal sites where the snow is often thin and patchy. However, as explained below, we have been able to study in separated efforts year-to-year changes in the BC concentration (by using ice cores collected) in the interior of the Antarctic Peninsula.

According to the reviewer’s suggestion, results from “Detaille Island” are shown separately in the new version of Fig. 2.

Comment of Reviewer 1

2) Once the uncertainty in the measurements is quantified, the uncertainty of BC induced snow albedo reduction needs to be addressed.

- When converting an absorption measurement on a filter to equivalent BC mass, a mass absorption coefficient (MAC) value must be prescribed (lines 397 – 425). The authors use a value obtained from the Monarch-71 soot standard. There are uncertainties around the use of this value, although the authors indicate the calculated BC concentration is really equivalent to the concentration of Monarch-71 soot (line 374) that would cause the measured absorption (eqn 1, line 365) by the particle laden filter. If this is the case, can you verify that the BC used in the albedo reduction model (Dang et al, supplemental info) is consistent with the MAC of Monarch 71 soot? These need to be the same or the measured concentrations used to feed the model would be inaccurate. The other option is to propagate inherent uncertainty in using a prescribed MAC through the calculations and add them to measurement uncertainties mentioned above. These uncertainties are critical given the very low BC concentrations reported in this study.

- On lines 219-220, the authors claim that 1-3 ng/g of BC can reduce albedo by 0.001 – 0.004. A brief mention of the model used and the reference to Dang et al. should be included in the main text as well as supplementary information).

Another good point that we are happy to clarify. Using optical properties of Monarch-71 soot as reference for pure BC is a standard practice for the MF technique application at least since Clarke et al., (1987) and yes, we confirm that the wavelength-dependent mass absorption coefficient (MAC) that we have used in our manuscript was also used in the albedo reduction model by Dang et al. (2015).

Moreover, please note that the BC concentrations at the 28 Antarctic sites where we surveyed are not particularly low (that is actually the main point of our manuscript). Thus, these concentrations are not exceptionally difficult to measure by using the MF technique; the very same technique has been used for measuring BC concentrations on the same order of magnitude from samples that included traces of dust and/or snow algae collected in Greenland (Doherty et al., 2010; Hegg et al., 2010), the Arctic (Dang et al., 2017; Clarke & Noone, 1985), and the Andes (Rowe et al., 2019).

According to the reviewer's suggestion, the model by Dang et al. (2015) for the albedo reduction, is now briefly mentioned in both the Main Text and the Supplementary Material.

Comment of Reviewer 1

3) The relative impact albedo has on ice melt is most significant when ambient temperature is near the melting point. When ambient temperature is significantly colder or warmer, the tiny changes in albedo due to the small concentrations of BC found in this study would likely be insignificant. The authors should include a discussion of ambient temperature on the melt results calculations.

Another excellent point made by the reviewer. We agree that the effects of the BC deposition on snow in the vast interior of Antarctica, where the temperature is well below the freezing point, are likely insignificant. That is why we only computed the BC-related snow melting at coastal sites where the summer temperature is much closer to 0°C; see Fig. 1a in Feron et al., (2021).

The positive forcing attributable to BC deposition accelerates seasonal snow melt as the amount of extra solar energy absorbed by the snowpack at BC-impacted sites can amount to several million joules (MJ). For example, the albedo reduction of 0.001-0.004 estimated for popular coastal tourist-landing sites in the Antarctic Peninsula makes the local snowpack absorb on average an extra of 2-8 MJ/m² over the 90-day summer period (when the shortwave solar irradiance in the region is on average about 250 W/m²). The enormous amount of extra energy absorbed by the snowpack will melt not only the snow at the zero-degree isotherm but also at sites where the air temperature is slightly below 0°C. There is sufficient energy to firstly make the snow temperature climb up to 0°C (which does not require a lot of energy) and to secondly melt the snow. Note that the specific heat capacity of ice is 2 kJ/kg/°C while the latent heat of fusion for melting the snow is 334 kJ/kg (Cohen, 1994).

According to the reviewer's suggestion, we have included this important discussion in the revised version of our manuscript.

Comment of Reviewer 1

4) The authors unequivocally state on lines 229 and 239 that BC is accelerating snow melt. While this might be true, they have not measured it or proven this to be the case. They need to make clear that they are hypothesizing snow is melting earlier due to BC concentrations.

We agree. According to the reviewer's suggestion, we have explicitly indicated in the revised version that, although our estimations of the BC-related snow melt are based on solid data and well-established science (see above), they are indeed estimations.

Comment of Reviewer 1

5) While the authors cite other studies showing long-range transport plays a minor role as a source of light absorbing aerosols in Antarctica, back trajectories specific to their samples should be calculated. The Antarctic peninsula is not that far from Argentina and downwind from Australia. One bad dust storm or smoke event could deposit enough light absorbing aerosol on the snow in a season to impact the entire study, even if it is short-lived and rare. The authors claim local activity is the primary source of light absorbing aerosol over a

wide area of Antarctica but have failed to back up this statement for their data specifically. If a seasonal analysis as recommended above were to reveal strong interannual variability at the sites, authors should look to see if continental influence could have played a role.

We agree. According to the reviewer's suggestion, we have included a new figure in the revised version showing clusters of 72-h backward trajectories for 3-month period before sampling. The 3-month period was selected since particles collect at the surface as the snow melts during summer (Conway et al., 1996).

Our back-trajectory analysis suggests that transport from Patagonia, for example, had only a minor role as a source of light-absorbing aerosols in Antarctica during the 3-month period prior to the sampling. These results are compatible with prior simulations of long-range BC transport from prominent sources in the Southern Hemisphere that have shown limited meridional airborne transport toward Antarctica (Pino et al, 2020; Neff & Bertler, 2015).

Although in separated efforts (see details below), we have detected episodic intercontinental transport of aerosols from wildfires in Patagonia, there has been no major wildfire in Patagonia since 2015 (more than a year before we began collecting snow samples).

Comment of Reviewer 1

6) The Union Glacier site while furthest from the peninsula does not have the lowest BC concentrations as the authors claim on line 171. Figure 2 shows that statistically the lowest BC concentrations were found at the Peterman and Detaille sites, identified as popular tourist sites with no permanent settlements.

We agree. We have reworded the text.

Comment of Reviewer 1

• Do any of the site measurements represent background Antarctica unaffected by local camps/activity?

No. Since prior efforts have already focused on clean sites, relatively far from research stations and tourist-landing sites (Kinase et al., 2020; Khan et al., 2018, 2019; Marquette et al., 2020; Arienzo et al., 2017; Bisiaux et al., 2012), this paper focuses on sites relatively close to research stations and popular tourist-landing sites.

However, please note that our team have indeed conducted separated efforts at sites far from local BC sources in Antarctica. These efforts based on ice and firm cores (see for example Hoffmann-Abdi et al., 2021) have confirmed that:

- the BC background concentration is consistently below 1 ng/g;
- massive wildfires in Patagonia can enhance (by more than one order of magnitude) BC deposition (last occurring in 2015; see Fig. 5 in Hoffmann-Abdi et al., 2021); and
- except for rare cases (like the wildfire in Patagonia in 2015), the year-to-year changes in the BC background concentration are minor.

These efforts provide strong evidence that the BC concentrations measured in our samples are, on the one hand, well above background levels and, on the other hand, mostly resulted from local emissions (and not from intercontinental transport).

Comment of Reviewer 1

• More details about the level of human activity at each site would help support the authors' claim that it is local activity and not continental influence that is the primary source of light absorbing aerosols.

We fully agree. According to the reviewer's suggestion, we have included further details in Table 1 on the facilities and local activities conducted at each of the 28 sites where we surveyed. Vessels, airplanes, diesel power plants, generators, helicopters, and trucks are all local BC-rich sources often used around the sampling sites.

Comment of Reviewer 1

In summary, the data are interesting and a more thorough analysis into the statistical significance, interannual variability in addition to inter-site variability, and an examination of potential sources would strengthen the findings considerably.

We thank the reviewer once again for her/his helpful suggestions and constructive remarks. As explained above, we have accordingly modified the attached version of our manuscript.

Best regards,

The authors.

References

Arienzo, M.M. *et al.* Holocene black carbon in Antarctica paralleled Southern Hemisphere climate. *J. Geophys. Res. Atmos.* **122**(13), 6713-6728 (2017).

Bisiaux, M. M. *et al.* Changes in black carbon deposition to Antarctica from two high-resolution ice core records, 1850-2000 AD. *Atmos. Chem. Phys.* **12**(9), 4107-4115 (2012).

Clarke, A. D., K. J. Noone, J. Heintzenberg, S. G. Warren, and D. S. Covert (1987), Aerosol light absorption measurement techniques: Analysis and intercomparisons, *Atmos. Environ.*, 21, 1455-1465

Clarke, A.D. & Noone, K.J. (2007). Soot in the Arctic snowpack: A cause for perturbations in radiative transfer. *Atmospheric Environment*, 41, 64-72.

Cohen, J. (1994). Snow cover and climate, *Weather* **49**, 150-156, <https://doi.org/10.1002/j.1477-8696.1994.tb05997.x>

Conway, H., Gades, A. & Raymond, C.F. Albedo of dirty snow during conditions of melt. *Water Resour. Res.* **32**(6), 1713-1718 (1996)

Dang, C., Brandt, R. E. & Warren, S. G. Parameterizations for Narrowband and Broadband Albedo of Pure Snow and Snow Containing Mineral Dust and BlackCarbon. *J. Geophys. Res. Atmos.* **120**(11), 5446-5468 (2015).

Dang, C., Warren, S.G., Fu, Q., Doherty, S.J., Sturm, M. & Su, J. (2017). Measurements of light-absorbing particles in snow across the Arctic, North America, and China: Effects on surface albedo. *Journal of Geophysical Research: Atmospheres*, 122(19), 10-149.

Doherty, S.J., Warren, S.G., Grenfell, T.C., Clarke, A.D. & Brandt, R.E. (2010). Light-absorbing impurities in Arctic snow. *Atmospheric Chemistry and Physics*, 10(23), 11647-11680.

- Doherty, S.J., Dang, C., Hegg, D.A., Zhang, R. & Warren, S.G. (2014). Black carbon and other light-absorbing particles in snow of central North America. *Journal of Geophysical Research: Atmospheres*, *119*(22), 12-807.
- Doherty, S.J., Hegg, D.A., Johnson, J.E., Quinn, P.K., Schwarz, J.P., Dang, C. & Warren, S.G. (2016). Causes of variability in light absorption by particles in snow at sites in Idaho and Utah. *Journal of Geophysical Research: Atmospheres*, *121*(9), 4751-4768.
- Feron S., Cordero R.R., Damiani A., Malhotra A., Seckemeyer, G., Llanillo P. Warming Events projected to become more frequent and last longer across Antarctica, *Scientific Reports*, *11*:19564 (2021).
- Grenfell, T. C. *et al.* Light Absorption From Particulate Impurities in Snow and Ice Determined by Spectrophotometric Analysis of Filters. *Appl. Opt.* **50**(14), 2037–48 (2011).
- Hegg, D.A., Warren, S.G., Grenfell, T.C., Doherty, S.J. & Clarke, A.D. Sources of light-absorbing aerosol in arctic snow and their seasonal variation. *Atmos. Chem. Phys.* **10**(22), 10923-10938 (2010).
- Hoffmann-Abdi, K., Fernandoy, F., Meyer, H., Freitag, J., Opel, T., McConnell, J. R., & Schneider, C. (2021). Short-Term Meteorological and Environmental Signals Recorded in a Firn Core from a High-Accumulation Site on Plateau Laclavere, Antarctic Peninsula. *Geosciences*, *11*(10), 428.
- Khan, A.L., Klein, A.G., Katich, J.M. & Xian, P. Local emissions and regional wildfires influence refractory black carbon observations near Palmer Station, Antarctica. *Front. Earth Sci.* **7**, 49 (2019).
- Khan, A.L. *et al.* Near-surface refractory black carbon observations in the atmosphere and snow in the McMurdo dry valleys, Antarctica, and potential impacts of Foehn winds. *J. Geophys. Res. Atmos.* *123*(5), 2877-2887 (2018).
- Kinase, T. *et al.* Concentrations and size distributions of black carbon in the surface snow of Eastern Antarctica in 2011. *J. Geophys. Res. Atmos.* **125**(1), p.e2019JD030737 (2020).
- Marquetto, L., Kaspari, S. & Simões, J.C. Refractory black carbon (rBC) variability in a 47-year West Antarctic snow and firn core. *The Cryosphere* **14**(5), 1537-1554 (2020).
- Neff, P. D. & Bertler, N. A. N. Trajectory modeling of modern dust transport to the Southern Ocean and Antarctica. *J. Geophys. Res. Atmos.* **120**, 9309–9322 (2015).
- Pino-Cortés, E. *et al.* The black carbon dispersion in the Southern Hemisphere and its transport and fate to Antarctica, an Anthropocene evidence for climate change policies. *Sci. Total Environ.* **778**, 146242 (2020).
- Rowe, P.M., Cordero, R.R., Warren, S.G., Stewart, E., Doherty, S.J., Pankow, A., Schrempf, M., Casassa, G., Carrasco, J., Pizarro, J. & MacDonell, S. (2019). Black carbon and other light-absorbing impurities in snow in the Chilean Andes. *Scientific Reports*, *9*(1), 1-16.
- Wang, X., Doherty, S.J. & Huang, J. (2013). Black carbon and other light-absorbing impurities in snow across Northern China. *Journal of Geophysical Research: Atmospheres*, *118*(3), 1471-1492.
- Warren, S.G. and Clarke, A.D. Soot in the atmosphere and snow surface of Antarctica. *J. Geophys. Res. Atmos.* **95**(D2), 1811-1816 (1990).

REVIEWERS' COMMENTS

Reviewer #1 (Remarks to the Author):

Thank you for thoroughly addressing all my concerns. I am satisfied with this version of the paper and support publication.

The authors provide important measurements of BC concentrations in snow and ice in Antarctica summer snow. Uncertainties are clearly shown and bring strength to relevancy of results. Methodology is well known, with an extensive history of publication by several groups and scientifically sound.

As written the work supports the conclusions.

I recommend publication.